# FLNeRF: 3D Facial Landmarks Estimation in Neural Radiance Fields

## Abstract

This paper presents the first significant work on directly predicting 3D face landmarks on neural radiance fields (NeRFs). This direct NeRF approach is shown to surpass existing single or multi-view image approaches. Our 3D coarse-to-fine Face Landmarks FLNeRF model efficiently samples from a given face NeRF individual facial features for accurate landmarks detection. Expression augmentation is applied at facial features in fine scale to simulate large emotions range including exaggerated facial expressions (e.g., cheek blowing, wide opening mouth) for training FLNeRF. Qualitative and quantitative comparison with related state-of-the-art 3D facial landmark estimation methods demonstrate the efficacy of FLNeRF, which contributes to downstream tasks such as high-quality face editing and swapping with direct control using our NeRF landmarks. Code and data will be available.

## 1 Introduction

3D facial landmarks prediction is fundamental in computer vision for various important applications. With the emergence of Neural Radiance Field (NeRF), a game-changing approach to 3D scene representation model for novel view synthesis, a 3D scene can be represented by a compact fully-connected neural network (Mildenhall et al., 2020). The network, directly trained on 2D images, is optimized to approximate a *continuous* scene representation function which maps 3D scene coordinates and 2D view direction to a view-dependent color and a density value. The implicit 5D continuous scene representation allows NeRF to represent more complex and subtle real-world scenes, overcoming reliance of explicit 3D data, where custom capture, sensor noise, large memory footprint, and discrete representations are long-standing issues. Further studies have improved the performance, efficiency and generalization of NeRF, with its variants quickly and widely adopted in dynamic scene reconstruction (Park et al., 2021; Xian et al., 2020; Li et al., 2020b; Pumarola et al., 2020; Du et al., 2021), novel scene composition (Ost et al., 2020; Yuan et al., 2021; Niemeyer & Geiger, 2020; Guo et al., 2020b; Liu et al., 2021; Yang et al., 2021; Müller et al., 2022; Kundu et al., 2022), articulated 3D shape reconstruction (Yang et al., 2022; Shao et al., 2022; Weng et al., 2022; Zhao et al., 2022; Jiang et al., 2022; Zheng et al., 2022; Xu et al., 2022; Chen et al., 2022; Noguchi et al., 2021; Peng et al., 2021) and various computer vision tasks, including *face NeRFs* (Gafni et al., 2021; Athar et al., 2022; Or-El et al., 2022; Sun et al., 2021; Deng et al., 2022; Hong et al., 2022), the focus of this paper.

This paper presents **FLNeRF**, which is to our knowledge the first work to accurately estimate 3D face landmarks directly on NeRFs. FLNeRF contributes a coarse-to-fine framework to predict 3D face landmarks directly on NeRFs, where keypoints are identified from the entire face region (coarse), followed by detailed keypoints on facial features such as eyebrows, cheekbones, and lips (fine). To encompass non-neutral, expressive and exaggerated expressions e.g., half-open mouth, closed eyes, and even smiling fish face, we apply effective expression augmentation and consequently, our augmented data consists of 110 expressions, including subtle as well as exaggerated expressions. This expressive facial data set will be made available. We demonstrate application of FLNeRF, by simply replacing the shape and expressions codes in (Zhuang et al., 2022) with our facial landmarks, to show how direct control using landmarks can achieve comparable or better results on face editing and swapping.

In summary, we propose FLNeRF, a coarse-to-fine 3D face landmark predictor on NeRFs, as the first significant model for 3D face landmark estimation directly on NeRF without any intermediate representations. We show this direct NeRF is significantly more accurate than state-of-the-art landmark detection from single or multi-view images. We demonstrate applications of accurate 3D landmarks produced by FLNeRF on multiple high-quality downstreamon tasks, such as face editing and face swapping (Figure 1).

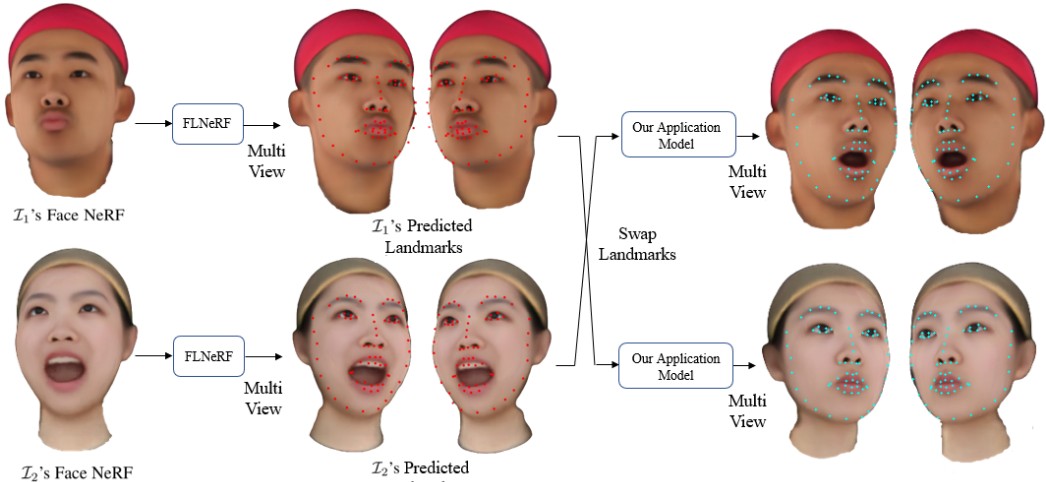

Figure 1: **Accurate 3D landmarks on face NeRF**. FLNeRF directly operates on dynamic NeRF, where an animator can easily edit, control and transfer emotion from another face NeRF. With precise landmarks on facial features, exaggerated facial expressions can be readily rigged and controlled by the animator.

Early models including Active Shape Models (ASM) (Milborrow & Nicolls, 2008; Cootes et al., 1995; Cootes & Taylor, 1992) and Active Appearance Models (AAM) (Cootes et al., 1998; Sauer et al., 2011) localize 2D landmarks on 2D images. However, 2D landmarks do not work well under large variations in pose and illumination. Moreover, applying 2D landmark prediction individually to multiple images capturing the same 3D face does not guarantee consistency. Even for single-image scenarios, some 3D face information (e.g., depth) is often estimated where 3D facial landmarks can be directly predicted. 3D landmarks estimation methods have been developed and applied in various downstream tasks, e.g., face recognition (Sharma & Kumar, 2021; Sharifisoraki et al., 2023; Mousavi et al., 2021), face synthesis (Zakharov et al., 2019), face alignment (Xia et al., 2022), and face reenactment (Kosarevych et al., 2020; Hao et al., 2020). Note that the input are still often 2D image(s) despite 3D *discrete* representations such as mesh, voxel, and point cloud are available, where controlled illumination, special sensors or synchronized cameras are required during data acquisition (Mildenhall et al., 2020; Pillai et al., 2019; Laszlo A. Jeni, 2019).

## 2 RELATED WORK

**2D Face Landmarks Prediction** ASM (Milborrow & Nicolls, 2008; Cootes et al., 1995; Cootes & Taylor, 1992) and AAM (Cootes et al., 1998; Sauer et al., 2011) are classic methods in 2D face landmarks prediction. Today CNN-based methods have become mainstream, consisting of heatmap regression models and coordinate regression models. Heatmap models (Wu et al., 2018; Zhu et al., 2019a; Sun et al., 2019; Valle et al., 2018) generate probability maps for each landmark location. However, face landmarks are not independent sparse points. Heatmap methods are prone to occlusion and appearance variations due to lack of face structural information. In contrast to heatmap regression models, directly learning landmarks coordinates could encompass weak structural knowledge (Trigeorgis et al., 2016b; Li et al., 2020a). Most coordinate regression methods (Sun et al., 2013; Trigeorgis et al., 2016a; Lv et al., 2017; Zhu et al., 2015; Su et al., 2019) progressively migrate predictions toward ground truth landmarks on 2D image.

**3D Face Model and 3D Face Landmarks Prediction** 3D Morphable Model (3DMM) (Blanz & Vetter, 1999) is among the earliest methods in representing 3D face which is usually used as an intermediate to guide learning of face models. However, this model restricts flexibility of face models due to its strong 3D prior and biased training data. To represent faces with wider range of expressions and preserve identity information, (Vlasic et al., 2006) proposed bilinear model, which

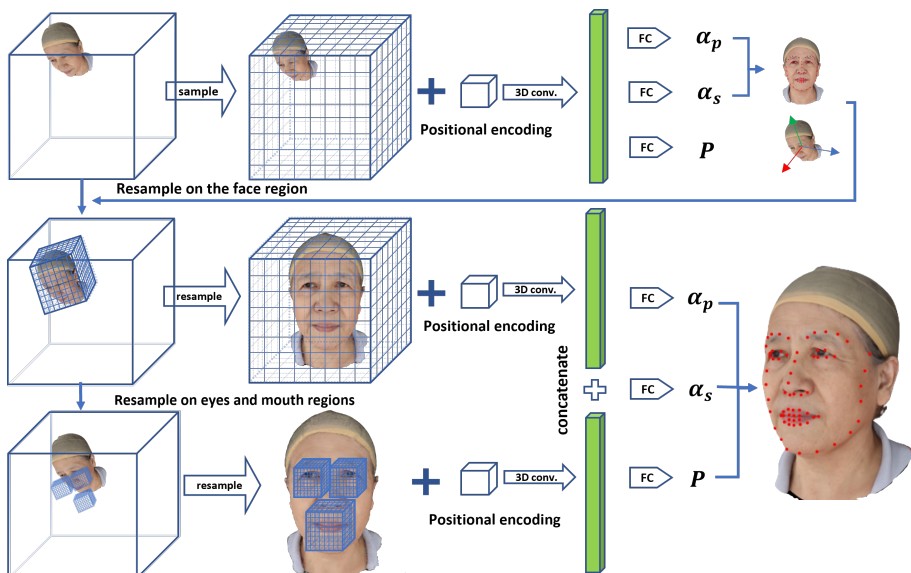

Figure 2: **FLNeRF pipeline of our 3D facial landmarks detection.** We choose 4 representative regions i.e., eyes, mouth and the whole face to detect facial landmarks. In each region, 4 channel volumes are sampled from the NeRF. Together with the 3D position encoding, feature volumes will be encoded as a 1D vector by 3D VolResNet or VGG backbone. Four separate 1D vectors received from the 4 coarse-to-fine scales (i.e., the whole face, left and right eyes including eyebrows, and lips including philtrum and cheekbone regions) are concatenated and decoded to bilinear parameters and pose (given by the transform matrix) using MLPs.

parameterizes face models in identity and expression dimensions. Facescape (Zhu et al., 2021) builds bilinear model from topologically uniformed models, through which 3D landmarks can be extracted, achieving better representation quality especially for identity preservation and wide range of expressions.

3D models in face-related tasks are more preferable to their 2D counterparts in representation power and robustness against pose and illumination changes. Before NeRFs, traditional 3D representation methods include voxel, mesh, and point cloud. However, building 3D models using these methods require controlled illumination, explicit 3D image, special sensors or synchronized cameras (Mildenhall et al., 2020; Pillai et al., 2019; Laszlo A. Jeni, 2019). Due to the demanding requirements for data acquisition, state-of-the-art 3D face landmarks prediction methods focus on localizing 3D landmarks on a single 2D image (Zhu et al., 2019b; Wu et al., 2021; Bulat & Tzimiropoulos, 2017; Feng et al., 2021; Guo et al., 2020a; Kumar et al., 2020; Yi et al., 2019). (Wu et al., 2021; Guo et al., 2020a; Zhu et al., 2019b; Yi et al., 2019) regress parameters of 3DMM followed by extracting 3D landmarks. (Bulat & Tzimiropoulos, 2017; Feng et al., 2021; Kumar et al., 2020) regress the coordinates of dense vertices or other 3D representations. These methods suffer from large memory footprint and long inference time, since they usually adopt heavy networks such as hourglass (Newell et al., 2016).

State-of-the-art 3D face landmarks localization methods have suboptimal accuracy and limited expression range and pose variations due to the 2D input. Although representations such as voxels and meshes can be constructed, an expressive face contains important subtle features which can easily get lost in such discrete representations. While low-resolution processing leads to severe information loss, high-resolution processing induces large memory footprint and long training and rendering time. Thus, a continuous, compact, and relatively easy-to-obtain 3D representation is preferred as input to 3D landmarks localization models for more direct and accurate estimation.

**Face NeRFs** Since NeRF represents 3D face continuously as solid (i.e., unlike point cloud crust surface), encoding 3D information in a compact set of weights (e.g., a $512\times512\times512$ voxel versus a $256\times256\times9$ network), and that it only requires multiview RGB images with camera poses, applying NeRF on face-related tasks has recently attracted research effort. (Gafni et al., 2021) combines a scene representation network with a low-dimensional morphable model, while (Athar et al., 2022) utilizes a deformation field and uses 3DMM as a deformation prior. In (Or-El et al., 2022), a NeRF-style volume renderer is used to generate high fidelity face images. In face editing and synthesis, training NeRF generators (Sun et al., 2021; Deng et al., 2022; Hong et al., 2022) reveals promising prospects for the inherently continuous 3D representation of the volume space, with drastic reduction of demanding memory and computational requirements of voxel-based methods. Mo-FaNeRF (Zhuang et al., 2022) encodes appearance, shape, and expression and directly synthesizes

Table 1: Quantitative comparison of FLNeRF and representative methods in average Wing loss. All values are multiplied by 10. Empty entries mean different landmarks definition on the corresponding regions.

| Method | Predictor | Average Wing Loss of All Expressions | | | Average Wing Loss of Exaggerated Expression | | | Avg. #Fail |
|---|---|---|---|---|---|---|---|---|
| | | Mouth | Eyes | Nose | Mouth | Eyes | Nose | |
| 2D Estimation + Triangulation | RSN | 5.15±0.57 | - | - | 4.80±0.53 | - | - | 0.00 |
| | RTMDet | 4.93±0.47 | - | 4.81±0.38 | 4.66±0.41 | - | 4.88±0.25 | 0.00 |
| | DarkPose | 4.94±0.51 | - | 5.15±0.46 | 4.47±0.61 | - | 5.43±0.55 | 0.00 |
| | DeepPose-SW | 4.86±0.60 | - | 5.02±0.46 | 4.49±0.47 | - | 5.38±0.51 | 0.00 |
| | STAR | 5.94±1.01 | 5.47±0.56 | 5.44±1.00 | 6.42±1.18 | 6.23±1.01 | 6.90±1.59 | 56.83 |
| | 2D FAN | 4.94±0.47 | 4.94±0.17 | 5.19±0.15 | 4.64±0.22 | 5.03±0.16 | 5.32±0.22 | 5.02 |
| | SPIGA | 5.61±0.98 | 5.13±0.66 | 5.21±0.92 | 5.79±0.93 | 5.84±0.94 | 6.33±1.28 | 57.99 |
| | PIPNet | 5.90±0.84 | 5.55±0.62 | 5.41±0.62 | 5.49±0.75 | 5.96±0.73 | 5.87±0.65 | 2.11 |
| Averaged 3D Estimation on Single Images | 3DDFA | 1.71±0.66 | 2.10±0.34 | 1.42±0.22 | 4.07±0.22 | 2.52±0.36 | 1.54±0.21 | 48.29 |
| | SynergyNet | 2.63±0.62 | 3.63±0.30 | 1.80±0.20 | 3.85±0.56 | 3.96±0.28 | 1.79±0.19 | 1.88 |
| | 3D FAN | 2.43±0.61 | 3.60±0.37 | 1.94±0.23 | 3.58±0.47 | 4.25±0.45 | 1.80±0.28 | 2.00 |
| | DECA | 2.22±0.58 | 3.25±0.28 | 1.29±0.21 | 3.46±0.48 | 3.85±0.33 | 1.48±0.11 | 0.00 |
| **Estimation on NeRF** | **FLNeRF** | **0.85±0.37** | **0.62±0.16** | **0.55±0.20** | **0.78±0.20** | **0.61±0.11** | **0.57±0.22** | - |

photo-realistic face. Continuous face morphing can be achieved by interpolating the three codes. We modify MoFaNeRF to support high-quality face editing and face swapping to demonstrate the advantages of direct control using 3D landmarks. (Gao et al., 2020; Rebain et al., 2022; Shi et al., 2022; Chan et al., 2022) reconstruct face NeRFs from a single image. We will show our FLNeRF can be generalized to estimate 3D face landmarks on 2D in-the-wild images, using face NeRFs reconstructed by EG3D Inversion (Chan et al., 2022).

# 3 3D FACE NeRF LANDMARKS DETECTION

Figure 2 shows the pipeline of FLNeRF which is a multi-scale coarse-to-fine 3D face landmarks predictor on NeRF. Our coarse model takes a face NeRF as input, and produces rough parameters estimation of the bilinear model, location and orientation (Sec. 3.1) of the input face by 3D convolution of the sampled face NeRF with position encoding. Unlike SynergyNet (Wu et al., 2021) which crops faces in 2D images, our coarse model can localize the pertinent 3D head in the NeRF space. Based on the estimated coarse landmarks, our fine model resamples from four regions: whole *face*, the left and right *eyes* including eyebrows, and *mouth* including lips, philtrum and cheekbone regions. In our coarse-to-fine implementation, the resolution of the sampled 3D volumes (coarse and fine) are respectively $64^3$.

The resampled volumes are then used to estimate more accurate bilinear model parameters with position encoding in Sec. 3.1. After describing how to benefit from the underlying continuous NeRF representation in sampling in Sec. 3.2, we will explain our coarse model in Section 3.3 and fine model in Section 3.4. Since there are only 20 discrete expressions in FaceScape (Zhu et al., 2021) with fixed head location and orientation, more diverse expressions and head poses are not covered in the dataset. To alleviate this limitation, we apply data augmentation to enrich our dataset to 110 expressions with different head poses per person, allowing our model to accurately locate and predict landmarks for faces with more complex expressions. We will describe our coarse data augmentation and fine expressions augmentation in Section 4.

## 3.1 BILINEAR MODEL

We utilize the bilinear model to approximate face geometry. FaceScape builds the bilinear model from generated blendshapes in the space of 26317 vertices $\times$ 52 expressions $\times$ 938 identities. Tucker decomposition decomposes the large rank-3 tensor into a small core tensor $C_r \in \mathbb{R}^{26317 \times 52 \times 50}$ and two low dimensional components $\mathbf{w}_{exp} \in \mathbb{R}^{52}$, $\mathbf{w}_{id} \in \mathbb{R}^{50}$ for expression and identity. Here, we only focus on the 68 landmarks subspace $C_r' \in \mathbb{R}^{68 \cdot 3 \times 52 \times 50}$. The flattened 68 3D landmarks $V_f \in \mathbb{R}^{3 \cdot 68}$ can be generated by Eq. (1):

$$V_f = C_r' \times \mathbf{w}_{id} \times \mathbf{w}_{exp} \tag{1}$$

To align $V_f$ with an input face NeRF, a transform matrix $P \in \mathbb{R}^{3 \times 4}$ is predicted. New aligned landmarks can be written as:

$$V_a = P \begin{bmatrix} V_f \\ 1 \end{bmatrix}. \tag{2}$$

## 3.2 NeRF SAMPLING

NeRF is a continuous representation underlying a 3D scene. In our case, each face NeRF is trained from 120 multi-view images in Facescape (Zhu et al., 2021). So far, most feature extractors are applied in discrete spaces such as voxel, mesh and point cloud, which inevitably induce information loss. In order to maximize the benefit of the continuous representation, we adopt a coarse-to-fine sampling strategy. Specifically, given a NeRF containing a human head, uniform coarse sampling will first be performed in the whole region of the NeRF with respect to the radiance and density channels to generate feature volumes (RGB is used to represent radiance). To make the radiance sampled on the face surface more accurate, we assume the viewing direction is looking at the frontal

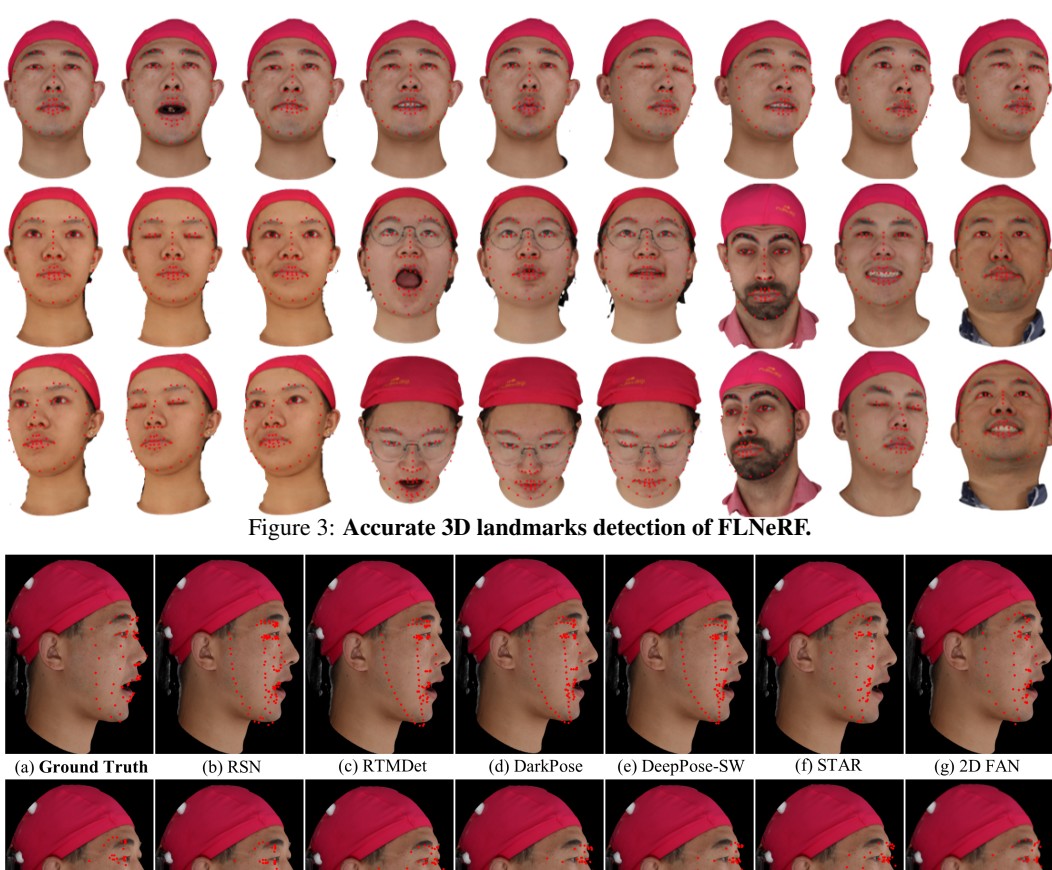

Figure 3: **Accurate 3D landmarks detection of FLNeRF.**

Figure 4: Qualitative comparison of FLNeRF with state-of-the-art methods, whose depth estimation and robustness under lateral view directions are erroneous. Each subfigure is estimated 3D landmarks overlayed on a lateral-view image, by triangulation or average single-image 3D landmarks estimation.

face when we sample the NeRF. We only utilize the radiance and density queried at given points of the NeRF, thus our model is applicable to most NeRF representations. To discard noisy samples, voxels with density smaller than a threshold (set to 20 by experiments) will be set to 0 in all channels (RGB and density), and voxel with density larger than the threshold will have the value one in density channel with RGB channels remaining the same. In the fine sampling, given the predicted coarse landmarks, orientation and translation of the head, the sampling regions of the whole face, eyes, and mouth are cubic boxes centered at the mean points of the landmarks belonging to corresponding regions with a suitable size proportional to the scale of the head. These cubic sampling boxes are aligned to the same rotation of the head. The same noise discarding strategy is used here.

### 3.3 COARSE MODEL

Inspired by the CoordConv (Liu et al., 2018), to enhance ability of 3D CNNs to represent spatial information, we add position encoding channels to each feature volume. Instead of directly using the Cartesian coordinates, a higher dimensional vector encoded from $x, y, z$ normalized to [0,1] are used as position encoding. The mapping function from $x, y, z$ to higher dimensional space is modified from that in (Mildenhall et al., 2020) which includes the original Cartesian coordinates:

$$\gamma(p) = (p, \sin(2^0\pi p), \cos(2^0\pi p), ..., \sin(2^{L-1}\pi p), \cos(2^{L-1}\pi p)). \tag{3}$$

We set $L = 4$ and $\gamma(\cdot)$ is applied to individual coordinates. We adopt the VoxResNet (Chen et al., 2016) and 3D convolution version of VGG (Simonyan & Zisserman, 2015) as our backbone to encode the pertinent feature volumes into a 1D long vector. Three seperate fully-connected layers are used as decoder to predict the transform matrix and bilinear model parameters. The transform matrix contains the head location and orientation. The Wing loss (Feng et al., 2018) is:

$$wing(x) = \begin{cases} \omega \ln(1 + |x' - x|/\epsilon) & \text{if } |x' - x| < \omega \\ |x' - x| - C & \text{otherwise} \end{cases} \tag{4}$$

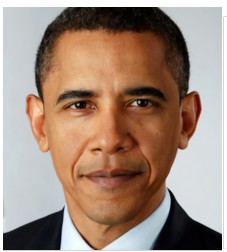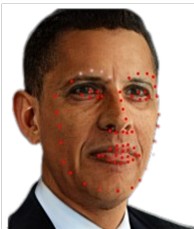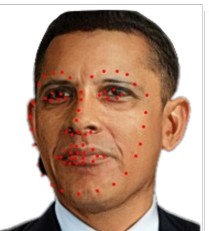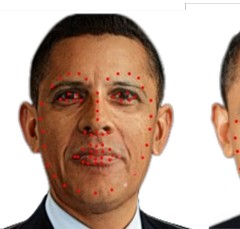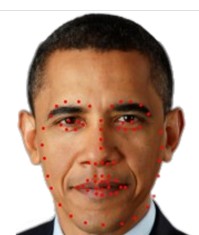

   **(a) In-The-Wild Image**      **(b) Prediction Overlayed on Side Views**      **(c) Front View**

Figure 5: FLNeRF can predict decent 3D landmarks on a suboptimal face NeRF reconstructed from a single in-the-wild image (Chan et al., 2022). (b) and (c) are overlayed side views and front view rendered from the face NeRF with predicted landmarks. (d) is the overlayed original image with predicted landmarks.

where we set $\omega = 10$ and $\epsilon = 2$. The $x' \in \mathbb{R}^{204 \times 1}$ is the predicted landmarks. The $x \in \mathbb{R}^{204 \times 1}$ is reshaped from ground truth landmarks $\in \mathbb{R}^{68 \times 3}$.

### 3.4 FINE MODEL

With the location, orientation, and coarse landmarks of the face given by the coarse model, a bounding box aligned with the head can be determined. Usually, the eyes and mouth have more expressive details. The bounding box of the eyes and mouth can also be determined, according to the coarse landmarks. Due to the low sampling resolution used in the coarse model and the inaccuracy of the coarse model prediction, the bounding boxes are made slightly larger to include all necessary facial features and their proximate regions. The same sampling method and position encoding as the coarse model is performed on these bounding boxes. Similar to the coarse model, VoxResNet and the 3D convolution version of VGG are used as the backbone to encode these four feature volumes into four 1D long vectors. These 1D long vectors containing expressive information on eyes, mouth, and the whole face are concatenated to predict the bilinear model parameters and a transform matrix, which are used to compute fine 3D landmarks. The loss function is the same as that in the coarse model.

### 3.5 EVALUATION AND COMPARISON

**Accuracy.** Fig. 3 shows qualitative results of our 3D landmark detection from NeRFs over a wide range of expressions. For quantitative evaluation and comparison with state-of-the-art methods, we randomly choose 5 identities as our test dataset. For the scale of prediction, we divide ground truth coordinates of 3D landmarks in Facescape (Zhu et al., 2021) by 100, and transform all predicted landmarks to the same coordinate system and scale of the divided ground truth. Table 1 shows quantitative comparison on all expressions (20 unaugmented expressions) and the exaggerated expression (unaugmented mouth stretching expression) of our FLNeRF with: (1) 2D face landmarks prediction on single images followed by triangulation, where state-of-the-art 2D predictors include RSN (Cai et al., 2020), RTMDet (Lyu et al., 2022), DarkPose (Zhang et al., 2020), DeepPose-SW (Zhu et al., 2020), STAR Zhou et al. (2023), 2D FAN (Bulat & Tzimiropoulos, 2017), SPIGA (Prados-Torreblanca et al., 2022), PIPNet (Jin et al., 2021); and (2) Averaged 3D face landmarks prediction on single images, where state-of-the-arts 3D predictors include 3DDFA (Zhu et al., 2019b), SynergyNet (Wu et al., 2021), 3D FAN (Bulat & Tzimiropoulos, 2017), DECA (Feng et al., 2021). The last column shows the average number of single images on which 2D or 3D landmarks estimators malfunction. The average number of estimations used to perform triangulation by (1), or to take average by (2) is thus (120 - Avg. #Fail). Since Facescape's annotations of landmarks on cheeks are different from all existing methods, we cannot compare performance on those landmarks quantitatively, while providing quantitative statistics on eyes, nose, and mouth. The last row of Table 2 shows performance of FLNeRF on all 110 expressions. Fig. 4 shows qualitative comparison, where superiority of our FLNeRF over all other methods on all landmarks could be clearly observed.

**Work for in-the-wild face NeRF?** To show FLNeRF is robust under various scenes and generalizable to in-the-wild face NeRF, we perform 3D face landmarks localization on face NeRFs reconstructed from a single in-the-wild face image using EG3D Inversion (Chan et al., 2022), which incorporates face localization and background removal, thus allowing FLNeRF to predict 3D landmarks on NeRFs containing only a face (and sparse noise). Fig. 5 shows that despite suboptimal reconstruction quality, FLNeRF can still accurately localize most feature points on the reconstructed face NeRF.

**Please refer to our supplementary material:** Section 1 provides additional training details. Section 2.1 offers a quantitative comparison in terms of additional metrics. Section 2.2 thoroughly analyzes the sub-optimal results obtained from 2D estimation followed by triangulation, supported by exhaustive experiments. Furthermore, Section 2.3 explains the reasons behind the inferior results of averaged 3D estimation on single images. Section 3 provides more visualization results.

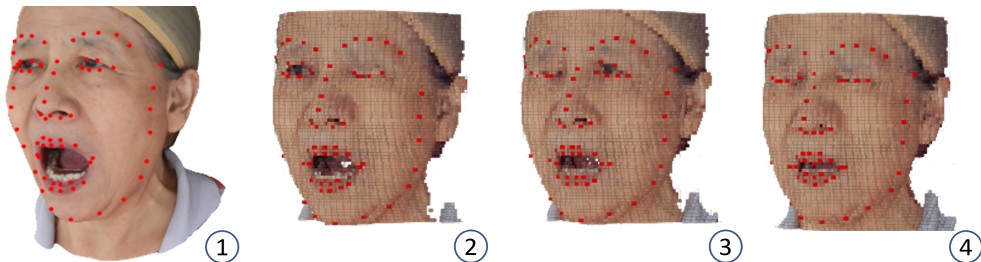

Figure 6: **Expression augmentation.** Subfigure 1 is the original stretch mouth NeRF with facial landmarks. Others are the feature volumes sampled non-uniformly from this NeRF using 3D TPS with the target landmarks.

## 4 AUGMENTATION AND ABLATION

### 4.1 DATA AUGMENTATION FOR COARSE MODEL

FaceScape consists of forward-looking faces situated at the origin. Taking into account NeRF implementations with different scales or coordinate systems, to boost generality and support 3D landmarks prediction on a wide variety of input NeRF containing a head, we augment the data set with various face locations, orientations and scales. We perform data augmentation during sampling these NeRFs into feature volumes $\mathbb{R}^{4 \times N \times N \times N}$ and assume the meaningful region of NeRF is within $[-1, 1]^3$, which can be easily normalized as such otherwise. Each sampled point $S \in [-1, 1]^3$ will be transformed by a matrix $\tau[\mathbf{R}t]$ to a new position, where $\mathbf{R} \in SO(3)$, $\tau \in [2, 3]$ and $t \in [-1, 1]^3$. New augmented feature volumes are generated by sampling NeRF at new sampling position. This operation is equivalent to scale, translate and rotate the head in the feature volumes. Although the sampled points may lie outside the captured NeRF, their densities are usually less than the threshold. Even some exceed the threshold, they are random noise points in the feature volumes that can be discarded by FLNeRF easily.

### 4.2 EXPRESSION AUGMENTATION FOR FINE MODEL

Fig. 6 illustrates our data augmentation to include more expressive facial features for training. First, we rig 20 expressions to 52 blendshapes based on FACS (Alkawaz et al., 2015). Then, we linearly interpolate these 52 blendshapes to 110 expressions. A total of 110 expression volumes from FaceScape (Zhu et al., 2021) are sampled non-uniformly from the given 20 expression NeRFs using 3D thin plate spline (3D TPS) (Bookstein, 1989). Note that the variation of the 20 discrete expressions in the FaceScape (Zhu et al., 2021) is insufficient for training 3D landmarks detector on the NeRF to cover wide range of facial emotions. Given a original $N$ 3D landmarks $\mathbf{L} \in \mathbb{R}^{N \times 3}$ and the target $N$ 3D landmarks $\mathbf{L}' \in \mathbb{R}^{N \times 3}$, we can construct $f(\mathbf{x})$ to warp $\mathbf{x} \in \mathbb{R}^3$ to $\mathbf{x}' \in \mathbb{R}^3$. Let $[\mathbf{l}_1, \mathbf{l}_2, \cdots, \mathbf{l}_{N-1}, \mathbf{l}_N]^\mathsf{T} = \mathbf{L}$ and $[\mathbf{l}'_1, \mathbf{l}'_2, \cdots, \mathbf{l}'_{N-1}, \mathbf{l}'_N]^\mathsf{T} = \mathbf{L}'$: $\mathbf{x}' = f(\mathbf{x}) =$

$$\mathbf{A}_0 + \mathbf{A}_1 \mathbf{x} + \sum_{i=1}^{N} \omega_{\mathbf{i}} U(\| \mathbf{l}_i - \mathbf{x} \|), \text{ where } \mathbf{A}_0 = \begin{bmatrix} a_x \\ a_y \\ a_z \end{bmatrix}, \mathbf{A}_1 = \begin{bmatrix} a_{xx} & a_{xy} & a_{xz} \\ a_{yx} & a_{yy} & a_{yz} \\ a_{zx} & a_{zy} & a_{zz} \end{bmatrix}, \omega_i = \begin{bmatrix} \omega_{ix} \\ \omega_{iy} \\ \omega_{iz} \end{bmatrix}.$$

$\mathbf{A}_0 + \mathbf{A}_1 \vec{x}$ is the best linear transformation mapping $\mathbf{L}$ to $\mathbf{L}'$. $U(\|\mathbf{x}_i - \mathbf{x}\|)$ measures the distance from $\mathbf{x}$ to control points $\mathbf{L}$. We use $U(r) = r^2 \log(r)$ as the radial basis kernel and $\|\cdot\|$ denotes $L_2$ norm. These coefficients $\mathbf{A}_0$, $\mathbf{A}_1$ and $\omega_i$ can be found by solving a linear system (supplemental material Section 4). In summary, a warped feature volume can be sampled non-uniformly from a NeRF by 3D TPS warp specified by the original and target landmarks. For each person in the FaceScape data set, a total of 110 expressions are available for training.

### 4.3 ABLATION STUDY

We conduct ablation on: (a) remove fine model, (b) remove expression augmentation, (c) use only two sampling scales, i.e., the first two rows in Fig. 2, (d) our full model.
Table 2 tabulates the ablation results using bilinear model and 3DMM respectively, where both of them use VoxResNet as backbone. It shows the advantage of bilinear model. Our full pipeline (d) achieves better performance than others, especially for the mouth region and exaggerated expressions. Ablation results using VGG as backbone can be found in Section 5 of our supplementary material.

## 5 APPLICATIONS

There has been no representative work on 3D facial NeRF landmarks detection that enables NeRF landmark-based applications, such as face swapping and expression editing, while producing realistic 3D results on par with ours. In this section, we will show how the landmarks estimated by FLNeRF can directly benefit MoFaNeRF (Zhuang et al., 2022).

Table 2: Since train/test data for coarse model only contains 20 basic expressions, we calculate the average Wing loss on these expressions for (a). For (b), (c) and (d), *whole face losses* are calculated on the test data set with 110 different expressions. *Mouth* and *Eyes* losses measure the corresponding landmarks' accuracy based on Wing loss. The last column shows results on basic mouth stretching expression and 10 augmented exaggerated expressions by method described in Section 4.2. All values are multiplied by 10.

|  | Average Wing Loss Using Bilinear Model | | | | Average Wing Loss Using 3DMM | | | |
|  | All Expressions | | | Exaggerated | Average Wing Loss of All Expressions | | | Exaggerated |
|  | Whole Face | Mouth | Eyes | Expressions | Whole Face | Mouth | Eyes | Expressions |
|---|---|---|---|---|---|---|---|---|
| (a) | 2.50±1.19 | - | - | - | 2.55±0.94 | - | - | - |
| (b) | 0.74±0.23 | 0.88±0.50 | 0.60±0.12 | 0.76±0.34 | 1.19±0.31 | 1.25±0.65 | 1.09±0.19 | 1.70±0.51 |
| (c) | 0.68±0.21 | 0.87±0.45 | 0.57±0.1 | 0.61±0.10 | 0.94±0.17 | 0.96±0.17 | 0.88±0.07 | 0.91±0.07 |
| **(d)** | **0.64±0.21** | **0.74±0.44** | **0.57±0.13** | **0.54±0.10** | **0.92±0.21** | **0.94±0.46** | **0.83±0.09** | **0.90±0.16** |

While MoFaNeRF generates SOTA results, we believe the range of expressive emotions is limited by its shape and expression code. 3D NeRF facial landmarks on the other hand provides *explicit* controls on facial expressions including fine and subtle details from exaggerated facial emotions. To directly benefit MoFaNeRF, we simply replace their shape and expression code with our 3D face landmarks location, which allows us to directly control NeRF's facial features and thus produce impressive results on morphable faces, face swapping and face editing. Please refer to Section 6.1 of the supplementary material for the model architecture of our modified MoFaNeRF, and Section 6.2 for ablation study.

## 5.1 FACE SWAPPING

We can swap the expressions of two identities by swapping their 3D landmarks. Two identities $\mathcal{I}_1$ and $\mathcal{I}_2$ may have different ways to perform the same expression. Feeding $\mathcal{I}_2$'s landmarks on a given facial expression with $\mathcal{I}_1$'s texture map to our modified MoFaNeRF enables $\mathcal{I}_1$ to perform the corresponding expression in $\mathcal{I}_2$'s way, the essence of face swapping. We show our modified MoFaNeRF can perform face swapping in Fig. 7, where the man takes on the woman's landmarks to produce the corresponding expression faithful to the woman's, and vice versa.

By simply appending the modified MoFaNeRF to FLNeRF, so as to perform downstream face swapping task after obtaining accurate prediction of 3D face landmarks, Fig. 1 shows that given two face NeRFs and their respective face landmarks, we can swap their expressions by simply swapping their face landmarks on NeRF.

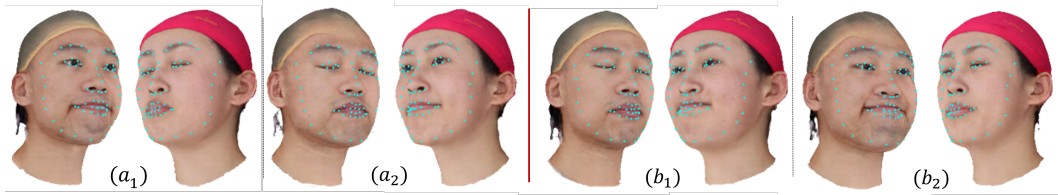

$(a_1)$ $(a_2)$ $(b_1)$ $(b_2)$

Figure 7: **Demonstration of face swapping by swapping landmarks.** $(a_1)(b_1)$ consist of rendered images from two different views by respectively feeding $\mathcal{I}_1$'s and $\mathcal{I}_2$'s texture map with ground truth landmarks to generate the pertinent NeRFs using our modified MoFaNeRF. $(a_2)(b_2)$ show images generated by respectively feeding to the network $\mathcal{I}_1$'s landmarks with $\mathcal{I}_2$'s texture map, and $\mathcal{I}_2$'s landmarks with $\mathcal{I}_1$'s texture map.

## 5.2 FACE EDITING

We can produce an identity's face with any expression given the corresponding landmarks and texture. Fig. 8 shows that our model can morph face by directly manipulating landmarks, where images on each row are rendered from NeRFs synthesized by linearly interpolating between the two corresponding NeRFs with landmarks of the leftmost expression and landmarks of the rightmost expression. Fig. 8 clearly demonstrates that our model can produce complex expressions even not included in our dataset. For example, middle images in the fifth row demonstrate our model's ability to represent a face with simultaneous chin raising and eye closing. Fig. 8 also shows that we can independently control eyes, eyebrows, mouth, and even some subtle facial muscles, with better disentanglement ability over MoFaNeRF (Zhuang et al., 2022) using shape and expression code. An extension of Fig. 8 could be found in the supplementary material.

We append our modified MoFaNeRF to FLNeRF, so as to perform downstream face editing after obtaining accurate prediction of 3D face landmarks. Fig. 9 shows that we can transfer one person's expression to another. In detail, we first obtain a face NeRF with the desired expression by feeding the corresponding landmarks into the modified MoFaNeRF. Then we apply our FLNeRF on the generated face NeRF to obtain accurate landmarks prediction. Finally, we use the predicted landmarks as input to the modified MoFaNeRF, together with texture map of another person, so that we obtain

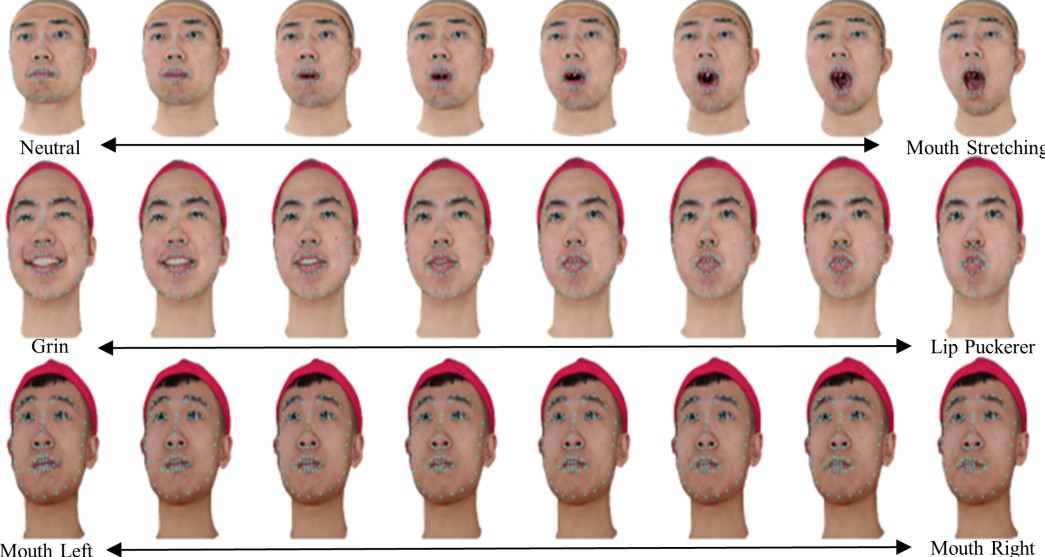

Figure 8: Demonstration of **face editing** via direct landmark control. For each row, images are rendered by interpolating landmarks of the left most expression and the right most expression. Figure 8 in the supplementary material is an entension of this figure.

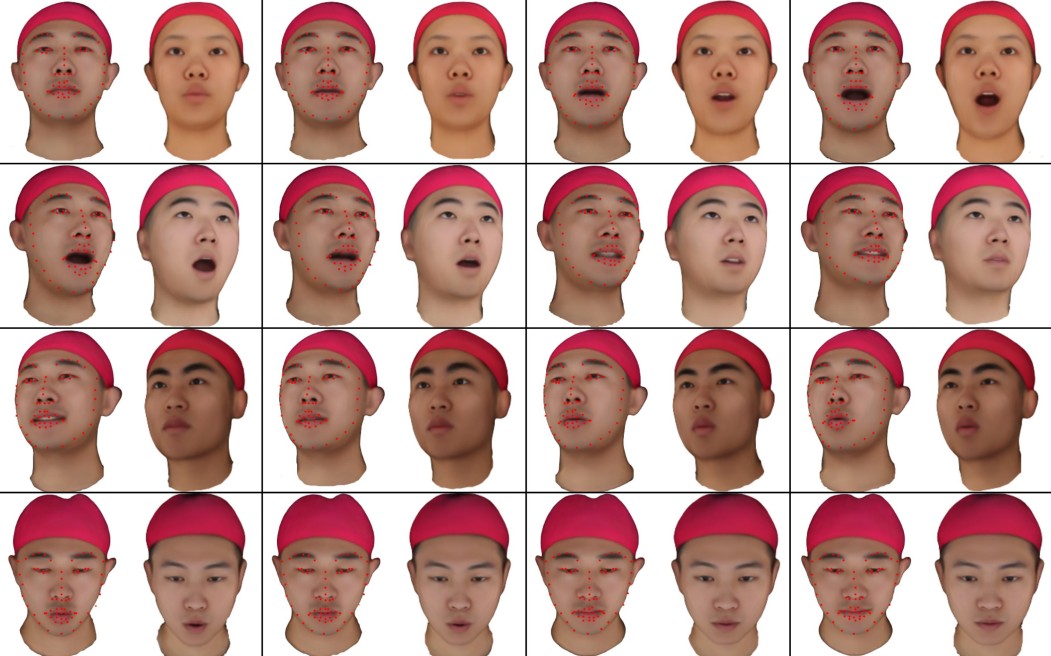

Figure 9: Demonstration of **expression transfer** by connecting the modified MoFaNeRF to FLNeRF as a downstream task. For each pair of images, the left face is the driver face, where landmarks on its NeRF are estimated by our FLNeRF and fed to our modified MoFaNeRF to drive the right face's expression.

face NeRF of another person with our desired expression. Refer to the supplemental video where face images are rendered from many viewpoints.

## 6 CONCLUDING REMARKS

We propose the first 3D coarse-to-fine face landmarks detector (FLNeRF) with multi-scale sampling that directly predicts accurate 3D landmarks on NeRF. Our FLNeRF is trained on augmented dataset with 110 discrete expressions generated by local and non-linear NeRF warp, which enables FLNeRF to give accurate landmarks prediction on a large number of complex expressions. We perform extensive quantitative and qualitative comparison, and demonstrate 3D landmark-based face swapping and editing applications. We hope FLNeRF will enable future works on more accurate and general 3D face landmarks detection on NeRF.

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
