# OpenReview forum: "FLNERF: 3D FACIAL LANDMARKS ESTIMATION IN NEURAL RADIANCE FIELDS"
_ICLR.cc/2024/Conference — Submitted to ICLR 2024_

### Official Review · Reviewer_Mp21 · 2023-10-30

**Soundness:** 3 good
**Presentation:** 3 good
**Contribution:** 3 good
**Rating:** 5
**Confidence:** 4

**Summary:**

In this paper, a facial landmark detection algorithm is proposed on the NeRF-generated 3D face images. The proposed method follows the coarse-to-fine approach. It first samples the 3D face image from face NeRF. It then detects the coarse facial landmark locations given the frontal face. Then, the fine model will refine the accurate landmark locations based on the estimated coarse locations from the previous step. The experimental results have been conducted to demonstrate the effectiveness of the proposed method.

**Strengths:**

It is interesting to see the facial landmark detection algorithm on the face NeRF images. The final detection accuracy seems to be significantly better than the other works.

**Weaknesses:**

The 3D facial landmark detection algorithm itself is straightforward. Once the 3D face is sampled from face NeRF, a typical detection algorithm is used.

=================================
After carefully reading the comments from the other reviewers. I do agree that the experiments are limited. Therefore, lower my rating.

**Questions:**

The authors should further justify the novelty of the proposed method.
It is not very clear how the feature and loss function are selected in the coarse model in Equations 3) and 4).
The authors should justify if the comparison of the proposed methods with other baseline methods in Table 1 is fair. Some of the baseline methods are only using 1 image for 3D detection. It is also not clear how the `triangulation` based method is done if detection is only from 1 2D image.
Is that possible to directly train/detect facial landmarks during face NeRF construction or to learn the mapping between face NeRF images and 3D landmark locations?

---

> ### Author Response · Authors · 2023-11-21
>
> We sincerely appreciate your insightful reviews and constructive suggestions. We are encouraged that you appreciate our method as the first to estimate 3D face landmarks estimation on NeRFs, surpassing state-of-the-art methods.
>
> For selection of feature and loss function in the coarse model in Eq. 3 and Eq 4, as highlighted by CoordConv, incorporating positional encoding can enhance the capacity of CNNs to extract spatial information effectively. Eq. 3 represents a widely utilized function that encodes positional information, which is subsequently appended to the volume features sampled from the NeRF. Eq. 4 refers to the Wing loss function, which has been shown to improve the training capabilities of deep neural networks in addressing small to medium-range errors, as compared to the $L_1$ and $L_2$ loss functions.
>
> For fairness of comparisons, we compared FLNeRF which estimates 3D face landmarks from face NeRFs, with triangulation and average single-image 3D landmarks estimation, as shown in Table 1 in main paper, and Table 1-5 in supplementary mateiral. For triangulation, we used state-of-the-art 2D face landmarks detection methods to detect 2D landmarks on single images, and then we performed triangulation to obtain 3D landmarks from 2D landmarks estimated from different single images. For average single-image 3D landmarks estimation, we performed 3D landmarks estimation on single images, and then take the average. We are not comparing our FLNeRF with methods performing 3D landmarks estimation on a single image. As elaborated in Sec. 3.5 in main paper, the number of images to perform 2D landmarks detection followed by triangulation, the number of images to perform average 3D single-image 3D landmarks estimation, and the number of images to train a face NeRF used by FLNeRF, are all the same. Therefore, our comparison is fair in terms of the number of input images.
>
> As pointed out in Sec. 3.5 in our main paper, definitions of landmarks on cheeks are indeed different from FaceScape. This conforms to R-1tLM's comment that landmarks in those 2D datasets around jaw do not have a fixed location, but represent the face occluding contour. However, when performing comparisons, as shown in Table 1 in our main paper, and Sec. 2.1, Sec. 2.2 and Sec. 2.3 (Table 1-5) in our supplementary material, we only show quantitative comparisons in those landmarks around mouth, eyes, and nose whose definitions are the same. Therefore, our comparison is fair in terms of landmarks definition.
>
> Quantitative and qualitative comparisons show that our FLNeRF which estimates 3D landmarks on NeRF significantly outperforms 2D landmarks estimation followed by triangulation, and average single-image 3D face landmarks estimation. Please refer to Sec. 2.2 and Sec. 2.3 in our supplementary material for reasons why our method produces better results. This could provide insights into the advantage of estimating 3D landmarks on NeRF over estimating 3D landmarks on images.
>
> NeRFs are becoming more and more popular these days. Even some mobile apps could reconstruct NeRF from mobile phone cameras. Currently those mobile apps require multi-view photo taking. Given that our FLNeRF could achieve reasonable results on low-quality NeRFs reconstructed from single images, we believe our technology could benefit users who could use their mobile apps to reconstruct high-quality in-the-wild NeRFs from multi-view images. Users could perform fancy downstream tasks with our technology, such as face animation shown in our paper, enabled by our FLNeRF + modified MoFaNeRF.
>
> Finally, we want to emphasize that we are the first to accurately estimate 3D face landmarks on NeRFs, with potential generalization to in-the-wild NeRFs. Despite the current scarcity of datasets with multi-view face images and ground truth 3D face landmarks, we are optimistic that our work will inspire further advancements in the community. Please let us know if you have further questions. We are happy to provide you detailed answers. We will appreciate it a lot if you could raise your score.

---

### Official Review · Reviewer_D4Fc · 2023-10-30

**Soundness:** 2 fair
**Presentation:** 3 good
**Contribution:** 2 fair
**Rating:** 5
**Confidence:** 3

**Summary:**

This paper presents a NERF based approach for predicting 3D face landmarks directly from neural radiance fields. This NeRF based solution is shown to surpass existing single or multi-view image approaches. The proposed 3D coarse-to-fine Face Landmarks FLNeRF model samples from a given face NeRF individual facial features for landmarks detection. Expression augmentation is applied at facial features in fine scale to simulate large emotions range including exaggerated facial expressions for training FLNeRF.

**Strengths:**

- The work presents a first contribution in using NERF for face landmarks detection in 3D.
- Results seems promising.
- The paper is presented in a good way.

**Weaknesses:**

- A NERF model is normally constructed for each 3D object to render. This limitation seems to apply also to this work. This represents quite a drawback for the proposed solution in that a different NERF model should be constructed for each identity. This drastically reduces the generality of the approach and results in a substantially increased computational effort that appears to be not much compatible with a problem of landmarks detection.
- Based on the above, the comparison with other methods that do not incur in such limitation is not completely fair in my opinion. Authors should at least clarify this point.
- In Section 3.5 it is reported that only five identities have been used in the test dataset. This appears as an insufficient number to derive a complete understanding of the proposed solution in comparison to state-of-the-art approaches. This very small number of identities does not have a sufficient statistical significance.
- It is not clear how much of the performance derive from data augmentation. A better insight of this should be provided.
- Limitations of the proposed method have not been discussed.

Minor corrections:
- Caption of Table 2: “by method described” --> by the method described

------
I have read the other reviews and the authors’ rebuttal, while also checked the revised paper. I thank the authors for the answers; however, I think the weaknesses of the work are still there. In my opinion, the experimental validation remains non satisfactory and the overall contribution below the ICLR standard. So, I keep my original rating.

**Questions:**

Q1: A NERF model is normally constructed for each 3D object to render. This limitation seems to apply also to this work. This represents quite a drawback for the proposed solution in that a different NERF model should be constructed for each identity. This drastically reduces the generality of the approach and results in a substantially increased computational effort that appears to be not much compatible with a problem of landmarks detection. Can author clarify this point?

Q2: Based on the above, could the authors present the results in a better way so that the comaprison account for other aspects than only accuracy?

Q3: It is not clear how much of the performance derive from data augmentation. A better insight of this should be provided.

---

> ### Author Response · Authors · 2023-11-21
>
> We sincerely appreciate your insightful reviews and constructive suggestions. We are encouraged that you appreciate our method as the first to estimate 3D face landmarks estimation on NeRFs, surpassing state-of-the-art methods.
>
> For Q1, we do need to train a seperate NeRF for each inference. However, instant-ngp already accelerated NeRF training from hours to seconds. Those 2D landmarks detections followed by triangulation or single-image 3D landmarks detections are also not immediate, taking from 30 seconds to 2 minutes. We believe as NeRF being more and more popular, more NeRF training acceleration techniques will be proposed. Also, there are many downstream tasks that do not require super fast face landmarks detection speed. For example, people may be able to produce his/her own face animations enabled by our FLNeRF + modified MoFaNeRF, as shown in our paper and video. Many mobile apps could already reconstruct NeRFs using mobile phone cameras in seconds. Imagine people could possibly make vivid face animations totally on their own, driving a cartoon face, a super man's face, or a celebrity's face by simply performing corresponding expressions themselves and feed the face NeRFs to our FLNeRF + modified MoFaNeRF pipeline, without experts drawing every frame.
>
> For Q2, please refer to Sec. 2.1, Sec. 2.2 and Sec. 2.3 (Table 1-5) in our supplementary material. They provide detailed analysis of why performing 3D landmarks estimation on NeRFs is a better choice. Our estimation process can be effectively performed in a single forward pass through the model. As a result, the inference time for each expression is less than one second (tested on the GTX 1080), which is comparable to the magnitude of the 2D landmarks detector.
>
> For Q3, for the improvement derived from the data augmentation, please refer to Sec. 4.3 in the main paper. This section presents an ablation study, illustrating the effects of excluding expression augmentation.
>
> Finally, we want to emphasize that we are the first to accurately estimate 3D face landmarks on NeRFs, with potential generalization to in-the-wild NeRFs. Despite the current scarcity of datasets with multi-view face images and ground truth 3D face landmarks, we are optimistic that our work will inspire further advancements in the community. Please let us know if you have further questions. We are happy to provide you detailed answers. We will appreciate it a lot if you could raise your score.

---

### Official Review · Reviewer_1tLM · 2023-10-31

**Soundness:** 2 fair
**Presentation:** 3 good
**Contribution:** 2 fair
**Rating:** 3
**Confidence:** 5

**Summary:**

The paper presents a model, termed FLNeRF, for estimating 3D facial landmarks from a face NeRF representation. In a first step it performs a coarse sampling of the NeRF volume to obtain an initial estimate of the face parameters. In a second step it re-samples again, at a finer scale, the face, eyes and mouth spatial locations. In both steps the reconstructed face volume, combined with positional encoding,  are the input to a CNN that estimates the face pose and the configuration parameters of a bi-linear model (a compressed version of FaceScape's) representing identity and expression, from which a set of 3D landmarks can be produced.

The experimentation evaluates the accuracy of the estimated landmarks so as their use for face editing and swapping.

**Strengths:**

The paper is easy to read and the problem addressed, face landmark estimation, is quite significant, with relevant applications and theoretical issues.

**Weaknesses:**

As the abstract reads, the central claim in the paper is that the approach can accurately estimate 3D face landmarks surpassing existing single or multi-view approaches. Also, since the starting point of the approach is a NeRF, to stress its practical use, the last sentence in section 2 reads "We will show our FLNeRF can be generalized to estimate 3D face landmarks on 2D in-the-wild images, using face NeRFs reconstructed by EG3D Inversion."

The paper does not convincingly demonstrate any of these claims.

The proposed approach is compared with the reconstructions computed with the landmarks detected with 2D methods and the 3D landmarks obtained with 3D methods in terms of the average Wing loss values multiplied by 10. I have several comments concerning this experiment:
- The evaluation is performed with 5 identities from the test dataset from FaceScape.  While this experiment provides some information, a sound comparison should include several other benchmark datasets in the literature and a widely used metric, such as e.g. the mean, median and std reconstruction errors.
- Is the comparison fair?  I have doubts since FLNeRF was trained with a train set from FaceScape, whereas competing approaches seem to have been trained with different datasets.
- In 2D datasets landmarks around the jaw do not have a fixed location, but rather represent the face occluding contour, so a reconstruction from their correspondences does not make much sense.

Finally, the experimentation concerning the estimation of landmarks on in-the-wild images, was made with a single image of president Obama, in which the estimated landmark locations are not very good.

**Questions:**

The authors should elaborate more on the complexity of estimating a detailed NeRF from an image, compared to a set of facial landmarks, and what are the advantages of estimating the landmarks from the NeRF, rather than from the image.

Specific questions:
- Is the comparison in Table 1 fair?
- Does a test on a single image provide sufficient experimental support to conclude that FLNeRF can be generalized to estimate 3D face landmarks on 2D in-the-wild images?

---

> ### Author Response · Authors · 2023-11-20
>
> We sincerely appreciate your insightful reviews and constructive suggestions. We are encouraged that you appreciate the importance of 3D face landmarks estimation and our method to solve the problem.
>
> For profoundness of comparisons, we also provided quantitative results in terms of adaptive average Wing loss and MSE. Please refer to Sec. 2.1 and Sec. 2.2 in the supplementary material, which show comparison in terms of more metrics in Table 1, Table 2, Table 3, Table 4 and Table 5 in the supplementary material. We conducted these comparisons using 20 different expressions for each identity, resulting in 100 diverse test cases. Acknowledging your insightful suggestion, we agree that testing on additional datasets would be beneficial. However, since NeRF and its variants are still young, and few people recognized its potential in 3D face landmarks estimation, FaceScape remains the only dataset to our knowledge providing multi-view face images with ground truth 3D face landmarks. Nonetheless, our tests on face NeRFs reconstructed by EG3D from single in-the-wild images demonstrate potential applicability to other datasets from another perspective. We believe that our work could stimulate the field of 3D face landmarks estimation by bringing the insights that estimating 3D landmarks on NeRF outperforms triangulation and average single-image 3D face landmarks estimation. Hopefully with this insight, more datasets with multi-view face images and 3D ground truth landmarks will be constructed.
>
> For fairness of comparisons, we compared FLNeRF which estimates 3D face landmarks from face NeRFs, with triangulation and average single-image 3D landmarks estimation, as shown in Table 1 in main paper, and Table 1-5 in supplementary mateiral. For triangulation, we used state-of-the-art 2D face landmarks detection methods to detect 2D landmarks on single images, and then we performed triangulation to obtain 3D landmakrs. For average single-image 3D landmarks estimation, we performed 3D landmarks estimation on single images, and then take the average. As elaborated in Sec. 3.5 in main paper, the number of images to perform 2D landmarks detection followed by triangulation, the number of images to perform average 3D single-image 3D landmarks estimation, and the number of images to train a face NeRF used by FLNeRF, are all the same. Therefore, our comparison is fair in terms of the number of input images.
>
> As pointed out in Sec. 3.5 in our main paper, definitions of landmarks on cheeks are indeed different from FaceScape. This conforms to your comment that landmarks in those 2D datasets around jaw do not have a fixed location, but represent the face occluding contour. However, when performing comparisons, as shown in Table 1 in our main paper, and Sec. 2.1 and Sec. 2.2 (Table 1-5) in our supplementary material, we only show quantitative comparisons in those landmarks around mouth, eyes, and nose whose definitions are the same. Therefore, our comparison is fair in terms of landmarks definition.
>
> Quantitative and qualitative comparisons show that our FLNeRF which estimates 3D landmarks on NeRF significantly outperforms 2D landmarks estimation followed by triangulation, and average single-image 3D face landmarks estimation. Please refer to Sec. 2.2 and Sec. 2.3 in our supplementary material for reasons why our method produces better results. Hope this could answer your question regarding the advantage of estimating 3D landmarks on NeRF over estimating 3D landmarks on images.
>
> We are grateful for your suggestions on generalization. Accordingly, we will revise our terminology to 'generalization to in-the-wild NeRF' for greater accuracy. The reason for showing estimations results by our FLNeRF on face NeRFs reconstructed by EG3D from single images is to show that our FLNeRF could produce reasonable results even given face NeRFs of low quality and background. NeRFs are becoming more and more popular these days. Even some mobile apps could reconstruct NeRF from mobile phone cameras. Currently those mobile apps require multi-view photo taking. Given that our FLNeRF could achieve reasonable results on low-quality NeRFs reconstructed from single images, we believe our technology could benefit users who could use their mobile apps to reconstruct high-quality in-the-wild NeRFs from multi-view images. Users could perform fancy downstream tasks with our technology, such as face animation shown in our paper, enabled by our FLNeRF + modified MoFaNeRF.
>
> Finally, we want to emphasize that we are the first to accurately estimate 3D face landmarks on NeRFs, with potential generalization to in-the-wild NeRFs. Despite the current scarcity of datasets with multi-view face images and ground truth 3D face landmarks, we are optimistic that our work will inspire further advancements in the community. Please let us know if you have further questions. We are happy to provide you detailed answers. We will appreciate it a lot if you could raise your scores.

---

### Meta-Review · Area_Chair_Jk8K · 2023-12-12

**Metareview:**

The paper presents the first work on directly estimating 3d face landmarks on NeRFs. The method utilizes a coarse-to-fine estimation strategy. In both scales, the reconstructed face volume and positional encoding are the input to a CNN to estimate the face pose and the parameters of a bilinear model, from which the 3d landmarks can be calculated.
Strengths: first work in using NeRF for 3d face landmark detection
Weaknesses: (1) it's not verified that the method can be generalized to in-the-wild images. (2) limited evaluation. (3) weak motivation

The missing parts include (1) persuasive evaluations to show the advantages of the method, (2) comparison to alternative methods like using a typical detection algorithm on a 3d face sampled from NeRF, and (3) enhancing the motivation.

**Justification For Why Not Higher Score:**

Although the paper is the first to propose the problem and come up with a solution, the results are not persuasive hence the motivation is weak. It got negative feedbacks from all three reviewers.

**Justification For Why Not Lower Score:**

N/A

---

### Decision · Program_Chairs · 2024-01-16

Reject